# Specific Combinations of Inflammatory, Angiogenesis and Vascular Integrity Biomarkers Are Associated with Clinical Severity, Coma and Mortality in Beninese Children with Plasmodium Falciparum Malaria

**DOI:** 10.3390/diagnostics12020524

**Published:** 2022-02-18

**Authors:** Bernard Tornyigah, Samuel Odarkwei Blankson, Rafiou Adamou, Azizath Moussiliou, Lauriane Rietmeyer, Patrick Tettey, Liliane Dikroh, Bernard Addo, Helena Lamptey, Maroufou J. Alao, Annick Amoussou, Caroline Padounou, Christian Roussilhon, Sylvie Pons, Benedicta Ayiedu Mensah, Nicaise Tuikue Ndam, Rachida Tahar

**Affiliations:** 1Université de Paris, MERIT, IRD, 75006 Paris, France; btonyigah@hotmail.com (B.T.); blanco3j96@gmail.com (S.O.B.); lauriane.rietmeyer@gmail.com (L.R.); nicaise.ndam@ird.com (N.T.N.); 2Department of Immunology, Noguchi Memorial Institute for Medical Research, University of Ghana, Legon, Accra P.O. Box LG 581, Ghana; tpm227@yahoo.com (P.T.); lilybeatrice2017@gmail.com (L.D.); weberchrome2468@gmail.com (B.A.); hlamptey@noguchi.ug.edu.gh (H.L.); mensah.benedictaa@gmail.com (B.A.M.); 3Department of Parasitology, Noguchi Memorial Institute for Medical Research, University of Ghana, Legon, Accra P.O. Box LG 581, Ghana; 4Institut de Recherche Clinique du Benin (IRCB), Calavi, Benin; adamourafiou@gmail.com (R.A.); azizathmoussiliou@yahoo.fr (A.M.); 5Département de Pédiatrie, Hôpital Mère-Enfant la Lagune (CHUMEL), Cotonou, Benin; amomj@yahoo.fr; 6Service de Pédiatrie, Centre Hospitalo-Universitaire, Suruléré (CHU-Suruléré), Cotonou, Benin; marwar68@yahoo.fr; 7Centre Hospitalier Universitaire de l’Oueme/Plateau, Porto-Novo, Benin; carolinepadonou@yahoo.fr; 8Unité de Génétique Fonctionnelle des Maladies Infectieuses, Département Génomes et Génétique, Institut Pasteur, 28 Rue du Docteur Roux, 75015 Paris, France; christian.roussilhon@pasteur.fr; 9Laboratoire Commun de Recherche Hospices Civils de Lyon-BioMérieux, Centre Hospitalier Lyon-Sud, Bâtiment 3F, 165 chemin du Grand Revoyet, 69310 Pierre-Bénite, France; sylvie.pons@biomerieux.com

**Keywords:** cerebral malaria, biomarkers, sICAM-1, EPCR, PTX3, PCT, suPAR, sTie-2, Ang-2

## Abstract

Malaria-related deaths could be prevented if powerful diagnostic and reliable prognostic biomarkers were available to allow rapid prediction of the clinical severity allowing adequate treatment. Using quantitative ELISA, we assessed the plasma concentrations of Procalcitonin, Pentraxine-3, Ang-2, sTie-2, suPAR, sEPCR and sICAM-1 in a cohort of Beninese children with malaria to investigate their potential association with clinical manifestations of malaria. We found that all molecules showed higher levels in children with severe or cerebral malaria compared to those with uncomplicated malaria (*p*-value < 0.005). Plasma concentrations of Pentraxine-3, Procalcitonin, Ang-2 and the soluble receptors were significantly higher in children with coma as defined by a Blantyre Coma Score < 3 (*p* < 0.001 for Pentraxine-3, suPAR, and sTie-2, *p* = 0.004 for PCT, *p* = 0.005 for sICAM-1, *p* = 0.04 for Ang-2). Moreover, except for the PCT level, the concentrations of Pentraxine-3, suPAR, sEPCR, sICAM-1, sTie-2 and Ang-2 were higher among children who died from severe malaria compared to those who survived (*p* = 0.037, *p* = 0.035, *p* < 0.0001, *p*= 0.0008, *p* = 0.01 and *p* = 0.02, respectively). These findings indicate the ability of these molecules to accurately discriminate among clinical manifestations of malaria, thus, they might be potentially useful for the early prognostic of severe and fatal malaria, and to improve management of severe cases.

## 1. Introduction

Malaria continues to be a major public health challenge. In 2020, the annual global mortality reached 627,000, hence leading to a 12% increase compared to 2019 with the majority of the mortality occurring in Africa [1,2,3]. *Plasmodium falciparum* is the most widespread malaria agent in Africa, and the most virulent species responsible for severe and cerebral malaria. The pathogenicity of severe malaria is mainly attributable to the cytoadherence of infected erythrocytes (IEs) within microvascular endothelia of vital organs modulating properties resulting in immune and splenic evasion characterized by the different pathophysiology of the disease (reviewed in [4,5]). Additionally, host immune dysfunction and excessive endothelial activation have also been associated with malaria pathology [6,7]. Specifically, the types and levels of cytokines and chemokines produced by the host vascular cells (i.e., endothelial cells, platelets, T-cell lymphocytes, monocytes and macrophages) play a critical role in disease development by serving as either effectors or targets. Several cytokines are known to take part in the endothelial dysfunction associated with leucocytes and infected red blood cells sequestration via the induction of intercellular-adhesion-molecule-1 (ICAM-1) and endothelial-protein-C-receptor (EPCR) expression on the cell surface and also by modulation of their shedding in the blood circulation resulting in an increased level of soluble EPCR and ICAM-1 (sEPCR, sICAM-1) in children admitted to the hospital for severe malaria [4,6,7,8]. The expression of the membranous form of urokinase plasminogen activator receptor (uPAR) has been associated with lesions during cerebral malaria suggesting a role of molecules implicated in the plasminogen activation pathway in cerebral malaria including suPAR [9]. This molecule as well as the pattern recognition glycoprotein pentraxine3 (PTX3) were also found to differentiate between uncomplicated and severe malaria. These results, however, were obtained in a cohort of severe and cerebral malaria cases missing fatality [10].

In addition, angiopoietin 2 (Ang-2) and its tyrosine-kinase-receptor 2 (Tie-2) were found to be increased and associated with retinopathies in Malawian children with cerebral malaria. In combination with clinical parameters, Ang-2 improved mortality prediction among these children [11]. The results obtained so far highlight the interest to further investigate the potential of these molecules as reliable biomarkers for accurate prognosis of patients at risk of dying from severe malaria and therefore to improve case management.

In this study, we assessed in a cohort of 339 Beninese children presenting with various clinical manifestations of malaria, the level of four soluble receptors (suPAR, sEPCR, sICAM-1 and sTie-2 with its ligand protein Ang-2) and two other bioactive molecules including PTX3 and the calcitonin-precursor-hormone procalcitonin (PCT).

## 2. Materials and Methods

### 2.1. Study Design and Participants

This cross-sectional study was carried out from December 2017 to July 2019 in the south of Benin where children under 6 years consulting at hospitals including Centre-Hospitalier-Universitaire Mère-enfant, de la Lagune, Centre-Hospitalier-Universitaire of Suruléré, and Ménontin Hospital, in Cotonou, l’Oueme/Plateau Hospital in Porto-Novo, and Ouidah Hospital in Ouidah were enrolled. Porto-Novo is 41.2 km east of Cotonou, while Ouidah is 39 km west of Cotonou, both settings share a subtropical climate, with 2 rainy seasons (April to July and September to October) and 2 dry seasons. This makes the south of Benin a malaria holoendemic area with a mean entomological inoculation rate of 33 infective bites per person per year [12].

Children spontaneously presenting at the hospitals were recruited into the study if they presented a positive rapid diagnostic test for *P. falciparum* (DiaQuick-Malaria-*P. falciparum*-Cassette, Dialab; Hondastrasse, Austria) and met the World Health Organization clinical malaria definition criteria. All malaria cases were defined by a trained clinician as a microscopically confirmed *P. falciparum* mono infection. Children were grouped according to the clinical manifestations of malaria they presented at the time of admission to hospital. The cerebral malaria group (CM) had a Blantyre Coma Score (BCS) < 3, excluding any other cause of coma with or without other signs of severe malaria. The severe non-cerebral malaria group (SNCM) included children presenting with one or more of the following symptoms: pulmonary edema, acute respiratory distress syndrome, acute kidney failure, abnormal liver function, hemoglobinuria or severe anemia with a BCS > 2. The uncomplicated-malaria-group (UM) had *P. falciparum* parasitemia infection with fever, and either headache or myalgia without signs of life-threatening malaria or evidence of vital organ dysfunction and had a hemoglobin level above ≥5 g/dL. Children presenting other causes of morbidity in addition to malaria were excluded.

### 2.2. Blood and Data Collection

Up to 4 mL peripheral blood was collected into citrate-phosphate-dextrose-adenine containing tubes (BD Medical^®^ ZI des Iles-BP4-38801-Le Pont de Claix, France) from all participating children before treatment. Clinical, biological and demographic data of patients were captured in a questionnaire and entered in an ad hoc file for further analysis. All patients were treated according to the guidelines of the Beninese Ministry of Health.

### 2.3. Quantification of Plasma Bioactive Molecules

Plasma samples were assayed in duplicate blindly. Standard ELISA experiments were performed according to the manufacturer’s instructions (R & D Systems^®^, Minneapolis, MN, USA) for PTX3, Ang-2 and soluble receptors. Plasma samples were diluted to 1/4 for Ang-2, and sTie-2, 1/10 for PTX3, suPAR, and sICAM-1 and 1/20 for sEPCR. For PCT quantification, the one-step enzyme-linked fluorescent immunoassay (VIDAS BRAHMS bioMérieux, Lyon, France) was performed using 200 µL plasma. In all experiments, analyte concentrations were calculated according to standard curves obtained by the assessment of specific recombinant human proteins which systematically included negative control sera. The final results were expressed as ng/mL.

### 2.4. Statistical Analysis

All statistical analyses were conducted with Stata software v15 and GraphPad Prism v9. Continuous variables and categorical variables were compared between the three clinical groups (CM, SNCM and UM) using the Kruskal–Wallis test in conjunction with Dunn’s multiple comparison post-hoc tests and chi-square test, respectively. The median and IC 95 levels of analytes were compared between the groups, UM versus CM-SNCM, patients who survived versus those who died and BCS < 3 versus BCS > 2 among severe cases using the Mann–Whitney U-test and binary logistic regression for which the effect described was independent of covariates such as ethnicity, age, parasitemia and sex. 

The validity of each molecule as a biomarker for malaria-related severity, coma and fatality was assessed by a receiver operating characteristic (ROC) curve analysis, a two-dimensional measure of classification performance where the area under the ROC curve (AUC) accurately measures discrimination, i.e., reflects the power of a quantified parameter to distinguish between two clinical groups. The greater the AUC, the better the test [13].

## 3. Results

### 3.1. Cohort Description

Blood samples were collected from 339 children aged between 4 and 66 months in a prospective cohort study of children who came to the hospitals for consultation and who were subsequently admitted to the pediatric ward for intensive care or outpatient consulting for uncomplicated malaria. Among them, 234 had severe malaria, including 101 with cerebral malaria (CM) having coma with BCS < 3. Twenty-nine children also had impaired consciousness with BCS = 3 accompanied by other symptoms of severe malaria, while 104 children had severe malaria without impaired consciousness and were all ranked in the group of severe non-cerebral malaria (SNCM). Severe anemia was very common among children and was detected in 45.71% of children with SNCM and 42.57% of those with CM. Forty children (11.79%) died including 29 (28.71%) among the CM group and 11 (8.27%) among the SNCM group, with six of them having consciousness alteration with a BCS of 3, and five suffering from SNCM without neurological impairment with a BCS of 5. Most children died within 10–24 h after hospitalization. One hundred and five children suffered from UM: all were treated and recovered (Table 1).

### 3.2. Immune and Endothelial Activators Are Associated with Malaria-Related Severity, Coma and Mortality

Comparisons of Median levels between the clinical groups including UM versus SNCM or CM or SM versus CM showed that the levels of all tested analytes were higher in children with SNCM and CM compared to those with UM, all *p* < 0.005. However, with the exception of the sEPCR level, the concentrations of the other molecules were also significantly increased in children with CM defined as coma with a BCS < 3 compared to those among children with SNCM as defined by a BCS > 2. Furthermore, apart from the level of PCT which did not differ between children who died and those who survived, the levels of suPAR, sEPCR, sICAM-1, sTie-2, Ang-2 and PTX3 were higher in children who died from SNCM or CM than those who survived these pathologies (Figure 1, Figure 2 and Figure 3) A–G. Logistic regression analyses summarizing the odds ratios calculations are presented in Table 2.

In these analyses we defined disease severity as severe non-cerebral malaria and cerebral malaria versus uncomplicated malaria (CM-SNCM versus UM), presence versus absence of coma (defined by BCS < 3 versus BCS > 2) and mortality versus survival among severe cases, as outcome variables while the effect described was independent of covariates such as ethnicity, age, parasitemia and sex. The odds ratios show an increase of 51% for Ang-2 and 21% for sTie-2 in children with CM or SNCM compared to UM, indicating that SNCM was 1.51 and 1.21 times more likely to occur with a unit increase in Ang-2 and sTie-2, respectively. Only a slight increase in the odds (between 1.03 and 1.06) for suPAR, PCT and PTX3 was noticed with a statistically significant difference between the two groups of patients. Except for sICAM-1, for which the increase in the odds was close to zero (but still significant), the odds for sTie-2, PTX3 and suPAR levels were 1.09, 1.04, and 1.04 times higher in patients with CM compared to those with SNCM. There was also a 1.03, 1.0, and 1.03 times increase in the odds of suPAR, PTX3 and sEPCR, respectively, in fatal cases compared to survivors of CM or SNCM pathologies which were statistically significant.

### 3.3. Predictive Accuracy of Analytes

We compared the ability of the analytes to discriminate between the various clinical groups using ROC curves. Six out of the seven analytes showed good diagnosis performance for malaria severity, coma and mortality with PCT and sTie-2 having the highest areas under the curve with AUC values of 0.83 (95% CI, 0.75–0.91) for PCT and 0.78 (95% CI, 0.71–0.85) for sTie-2 (Figure 4). For coma, the highest diagnostic performance was shown by AUC values of 0.78 (95% CI, 0.72–0.85, *p* < 0.001) and 0.71 (0.58–0.83, *p* [0.01–0.001]) for PTX3 and sICAM-1, respectively (Figure 5), while the highest diagnostic performance for mortality was obtained for sEPCR and sICAM-1, both with an AUC of 0.77 (95% CI, 0.69–0.85, *p* < 0.001) and (95% CI, 0.64–0.89, *p* < 0.001), respectively (Figure 6). The three most predictive biomarkers for severity, coma and mortality are presented in Table 3, where they are classified according to the AUC value.

## 4. Discussion

Both the innate and adaptive immune responses depend critically on leucoytes migration across the vascular endothelium; therefore, factors associated with endothelial cell response including inflammation, tissue damage and injury aroused a high interest in studies of malaria patients. In this study, we assessed the level of seven markers of vascular and tissue integrity, angiogenesis and inflammation in the context of malaria. The interactions between these molecules are illustrated in Figure 7.

Our findings showed the potential of four soluble receptors including sEPCR, sICAM-1, suPAR and sTie-2 with its ligand Ang-2, and two other molecules PCT, and PTX3, as informative biomarkers of malaria disease severity, coma and mortality.

On admission, the levels of all the analytes considered were higher in children with CM or SNCM compared to children with UM as shown by a logistical regression analysis (Table 2) and non-parametric tests (Figure 1, Figure 2 and Figure 3). These results were validated by the AUC values generated from the ROC curve analysis. PCT and sTie-2/Ang-2 displayed the highest AUC values (0.83, 0.78 and 0.67, respectively), compared to the other markers (Figure 4 and Table 3). Besides, suPAR, sTie-2, Ang-2, sICAM-1, PCT and PTX3 were associated with coma, and PTX3 was the more promising biomarker of coma with an odd of 1.09 and an AUC of 0.78 (Figure 5). Regarding mortality, sEPCR was the best predictive biomarker, considering both logistic regression and AUC values, while sICAM-1 presented a high AUC value of 0.77 but was not significantly predictive according to the logistic regression analysis. However, there were lower AUC results with the other molecules suggesting weaker associations or a lower degree of dependence between mortality and the variables tested (Figure 6).

Interestingly, we found that the levels of suPAR, sICAM-1, sTie-2, Ang-2, PTX3 and PCT were associated with coma and that sEPCR, sICAM-1, suPAR, sTie-2, Ang-2 and PTX3 were associated with fatality. These molecules have also been found to be increased in patients with other infectious or chronical diseases such as sepsis, HIV-1-AIDS, tuberculosis, rheumatoid-arthritis and various cancers with suPAR, sEPCR and PTX3, often predicting poor clinical outcomes as in our present study [8,14,15,16].

sEPCR, sTie-2/Ang-2 and sICAM-1 are involved in cytoprotective activity to maintain endothelial barrier functions of microvessels. The soluble as well as the membranous form of EPCR bind with similar affinity to activated C-reactive-protein (APC) to maintain endothelium integrity through the activation of protease-activator-1 [17,18] promoting anticoagulation and anti-inflammation. During malaria, EPCR has been found to bind the P. falciparum erythrocyte-membrane-protein-1 (PfEMP1) to the same region as APC; therefore, it decreases the anticoagulant activity of APC and promotes thrombosis and obstruction of blood circulation [6,19]. Our data suggest that increased sEPCR levels detected in severe cases might be involved in maintaining APC anticoagulant activity in these patients, potentially preventing acute thrombosis.

Similarly, the level of Ang-2 and its receptor sTie-2 was higher in SNCM or CM groups as well as in coma and in fatal cases suggesting a pathophysiological association with CM. Interestingly, Ang-2 and the Ang-2/Ang-1 ratio were previously shown as independent predictors of metabolic acidosis, coma and mortality [11,20,21]. The Ang1/Ang-2-Tie-2 system is a paramount regulator of endothelial integrity with Ang-1 signals through Tie-2 to maintain vascular quiescence. This activity is antagonized by Ang-2 resulting in endothelial dysfunction and inflammation [22]. The sTie-2 contains the ligand binding domain that binds angiopoietins and inhibits their interaction with cellular Tie-2. Of note, sTie-2 is involved in regulating angiopoietins’ availability and its contribution is critical to vascular pathology when its level is impaired [23]. Therefore, the high production of sTie-2 may thwart the antagonizing effects of Ang-2 which was highly expressed in CM and SNCM during coma and in fatal cases. 

Moreover, the enhanced level of Ang-2 and sTie-2 was concomitant with the expression of sICAM-1 which was significantly higher in young Beninese patients with CM or SNCM as well as in children with coma and those who died (Figure 1, Figure 2 and Figure 3) C–E.

We also found that the suPAR level was significantly higher in children with SNCM or CM compared to those with UM, in children with BCS < 3 compared to those with BCS > 2 and in children who died from SNCM or CM making this molecule a potential marker of severity, coma and fatality (Figure 1, Figure 2 and Figure 3) A. Produced by activated monocytes and endothelial cells, suPAR plays several roles in innate immune defence and inflammation. It acts in the recruitment of effector cells (monocytes/macrophages) and in the clearance of pathogens at infection sites [24]. In addition to the uPA/suPAR binding which triggers the plasminogen activation system, suPAR also binds integrins and other receptors to activate different intracellular signaling pathways implicated in cell proliferation, invasion, angiogenesis and metastasis. suPAR has been described to be increased in cancers, in coronary heart diseases (CHD) and in infectious diseases including malaria [10,25,26,27]. An immunohistochemistry analysis of a CM patients brain showed an accumulation of suPAR in macrophages/microglial cells in Durks granuloma adjacent to petechial haemorrhages, as well as in astrocytes, and in endothelial cells [9]. However, the expression of suPAR was low in normal brains suggesting the association of suPAR expression with tissue damage of the blood brain barrier during CM [9]. A high suPAR level was associated with disease severity and poor prognosis in cancers, CHD and infectious diseases including SARS-CoV-2 [25,28,29,30]. 

With an AUC of 0.83, PCT was found to be the best biomarker to reveal the severity of malaria and this result is consistent with previous findings [31,32]. Interestingly, PCT was higher in children with BSC < 3 than those with BCS > 2 (*p* = 0.004) and showed moderate predictive value for coma with an AUC of [0.65], which is to our knowledge a novel finding. This molecule is able to discriminate between viral, bacterial, fungi or parasitic infections and is also used in guiding antibiotic treatments in patients [33]. Of note, PCT in combination with CRP, chitinase-3-like-protein and S100beta was recently described to be a promising biomarker in determining the presence, location and extent of traumatic intracranial lesions [34].

The level of PTX3 was higher in children with SNCM or CM compared to children with UM, in children with BCS < 3 compared to those with BCS >2 and in children who died compared to those who survived with significant diagnosis performance for severity, coma and mortality as shown by the result of the AUC analyses (Figure 4, Figure 5 and Figure 6). PTX3 is involved in the humoral innate immunity by recognizing microbial moieties and damaged tissues and in regulating inflammation and autoimmunity [35]. PTX3 is a pattern recognition molecule (PRM) expressed by both immune and non-immune cells upon stimulation by bacterial lipopolysaccharides and pro-inflammatory molecules such as IL-1 and TNFα [36,37]. Increased levels of plasma PTX3 were associated with the clearance of apoptotic cells by dendritic cells resulting from high immune response and endothelial dysfunction in cardiovascular disorders [38,39]. PTX3 was investigated as a biomarker in infectious diseases and has emerged as a powerful independent predictor of mortality as recently described in SARS-CoV2 syndrome [40]. However, the present study is only the second one that investigated PTX3 as a potential biomarker during malaria and revealed the potential of this molecule to distinguish SNCM and CM from UM cases, coma from impaired consciousness and conscious cases as well as fatal cases from patients who survived. This result highlights the potential interest of PTX3 as a biomarker with the capacity to assess different expressions of the disease and response to treatment in hospitalized malaria patients. 

Finally, the three most predictive molecules of severity, coma and mortality are presented in Table 3, and show that PCT had the highest predictive value for severity, while PTX3 presented the highest AUC value to predict coma and both sEPCR and sICAM-1 had the highest predictive value for fatality. These results strongly suggest that the assessment of a combination of these markers in the plasmas of patients will allow for evaluating properly the clinical state of the patients as well as the potential risk of death, in order to significantly improve case management.

## 5. Conclusions

Our findings show that in Beninese children with malaria, the levels of suPAR, sICAM-1, sEPCR, sTie-2, Ang-2, PTX3 and PCT differentiate children with SNCM and CM from children with UM, with PCT and sTie-2 providing the highest diagnosis performance for severity with an AUC of [0.83] and [0.78], respectively. Except for the sEPCR level, the levels of all the molecules tested were higher in children with BCS < 3 compared to children with BCS > 2, with PTX3 and sICAM-1 presenting the best diagnosis performance for coma with an AUC of [0.78] and [0.71], respectively. In addition, the sEPCR, sICAM-1, sTie-2, Ang-2, PTX3 and suPAR levels discriminated between children who died from SNCM and CM and those who survived these pathologies with the highest predictive value for sEPCR, and sICAM-1 with an AUC of [0.77]. 

Overall, the expression of these molecules during malaria, either at the onset of infection when triggering the inflammatory response such as suPAR and PTX3, or resulting later from an acute inflammatory response such as sICAM-1 and sEPCR, is associated with the pathogenesis of severe malaria in which the site of initiation of this response and its intensity may play a major role. Therefore, further research on the mechanisms of action of these molecules is needed to better understand their role in the pathogenesis of SNCM and CM and explore their potential as host-targeted therapeutics.

## Figures and Tables

**Figure 1 diagnostics-12-00524-f001:**
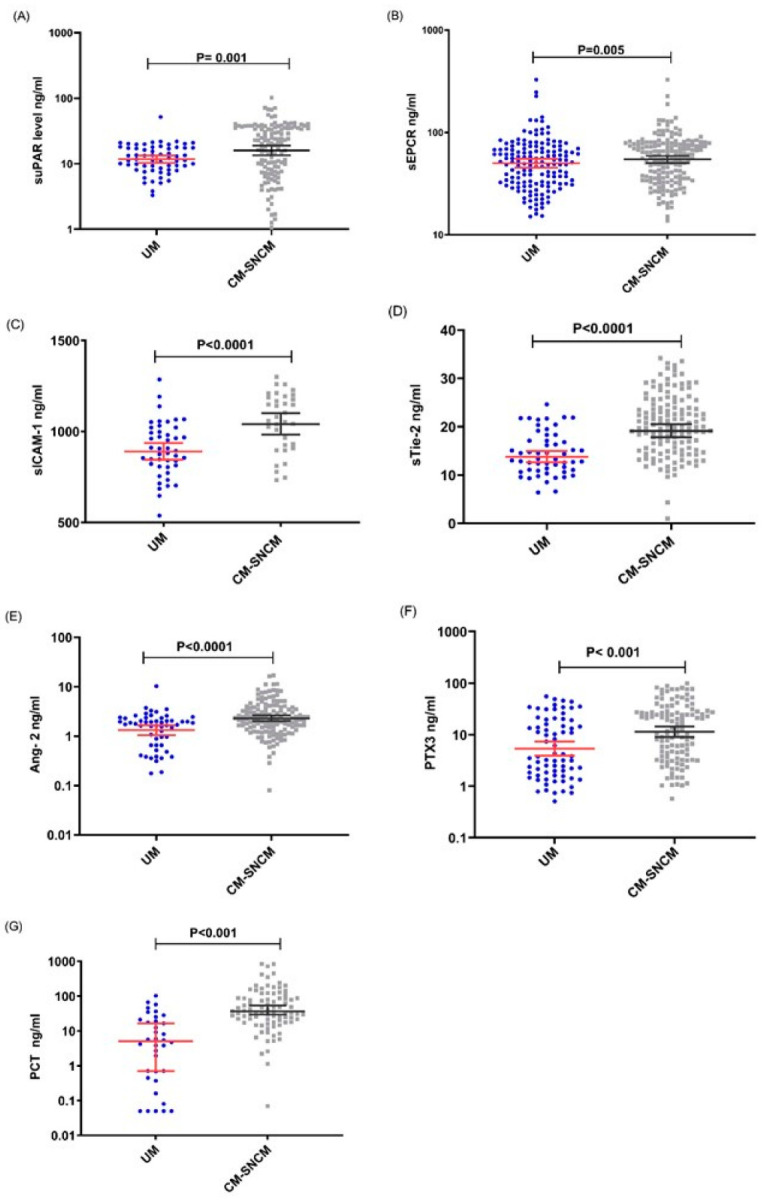
Plasma levels of suPAR, sEPCR, sICAM-1, sTie-2, Ang-2, PTX3 and PCT in children with uncomplicated malaria (UM) and in children with cerebral malaria or severe non-cerebral malaria (CM-SNCM). (**A**–**G**), Plasma levels of (**A**) suPAR, (**B**) sEPCR, (**C**) sICAM-1, (**D**) sTie-2, (**E**) Ang-2, (**F**) PTX3 and (**G**) PCT obtained in children with uncomplicated malaria (UM) (blue dots) and cerebral malaria or severe non-cerebral malaria (CM-SNCM) (gray squares) on the day of admission to hospital as measured by ELISA. Data are presented as dot plots with median with (95% CI). The Mann–Whitney U-test was performed for each comparison.

**Figure 2 diagnostics-12-00524-f002:**
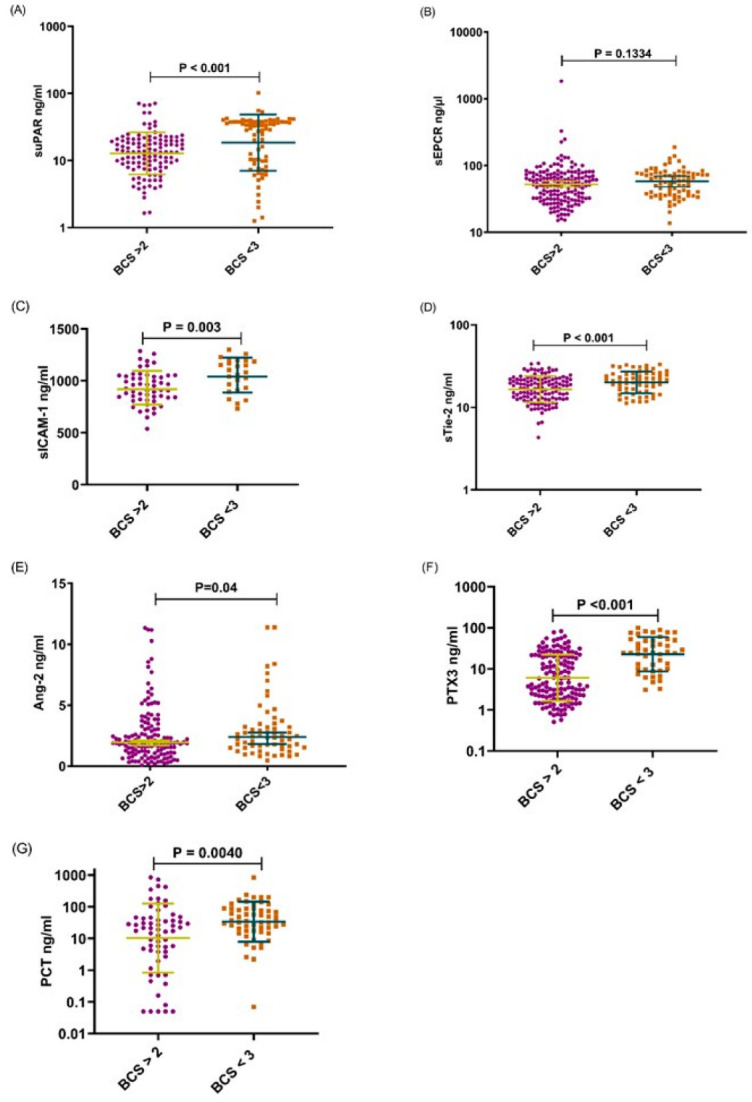
Plasma levels of suPAR, sEPCR, sICAM-1, sTie-2, Ang-2, PTX3 and PCT, in children with BCS < 3 compared to those with BCS > 2. (**A**–**G**), The levels of (**A**) suPAR, (**B**) sEPCR, (**C**) sICAM-1, (**D**) sTie-2, (**E**) Ang-2, (**F**) PTX3 and (**G**) PCT, in children with coma, i.e., with a BCS < 3 (brown squares) compared to those with a BCS > 2 (purple dots). The molecules were measured by ELISA on the day of admission to the hospital. Data are presented as dot plots with median and (95% CI). The Mann–Whitney U-test was performed for each comparison.

**Figure 3 diagnostics-12-00524-f003:**
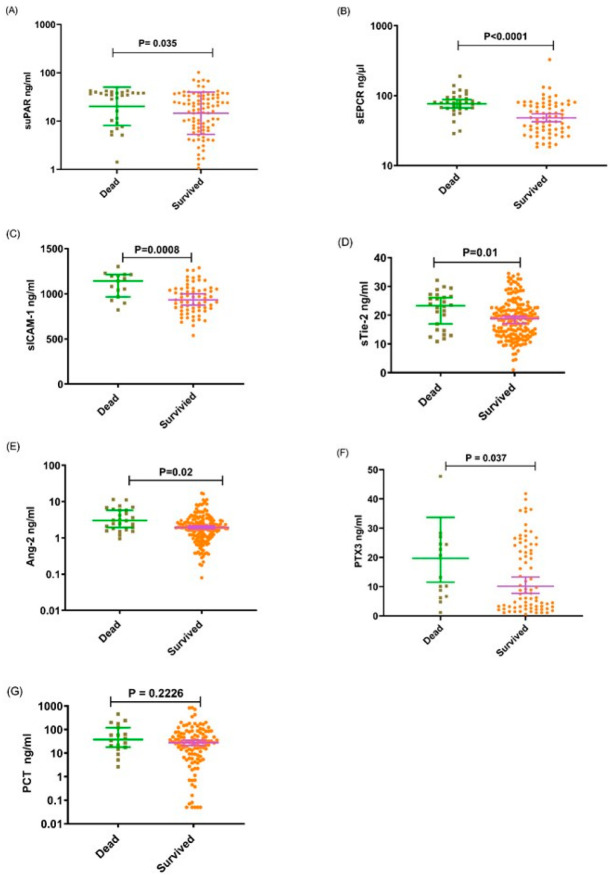
Plasma levels of suPAR, sEPCR, sICAM-1, sTie-2, Ang-2, PTX3 and PCT in children who survived severe non-cerebral malaria or cerebral malaria compared to those who died. (**A**–**G**), The levels of (**A**) suPAR, (**B**) sEPCR, (**C**) sICAM-1, (**D**) sTie-2, (**E**) Ang-2, (**F**) PTX3 and (**G**) PCT in children who survived severe or cerebral malaria (yellow dots) compared to those who died (gray squares) on the day of admission to hospital as measured by ELISA. Data are presented as dot plots with median and (95% CI). The Mann–Whitney U-test was performed for each comparison.

**Figure 4 diagnostics-12-00524-f004:**
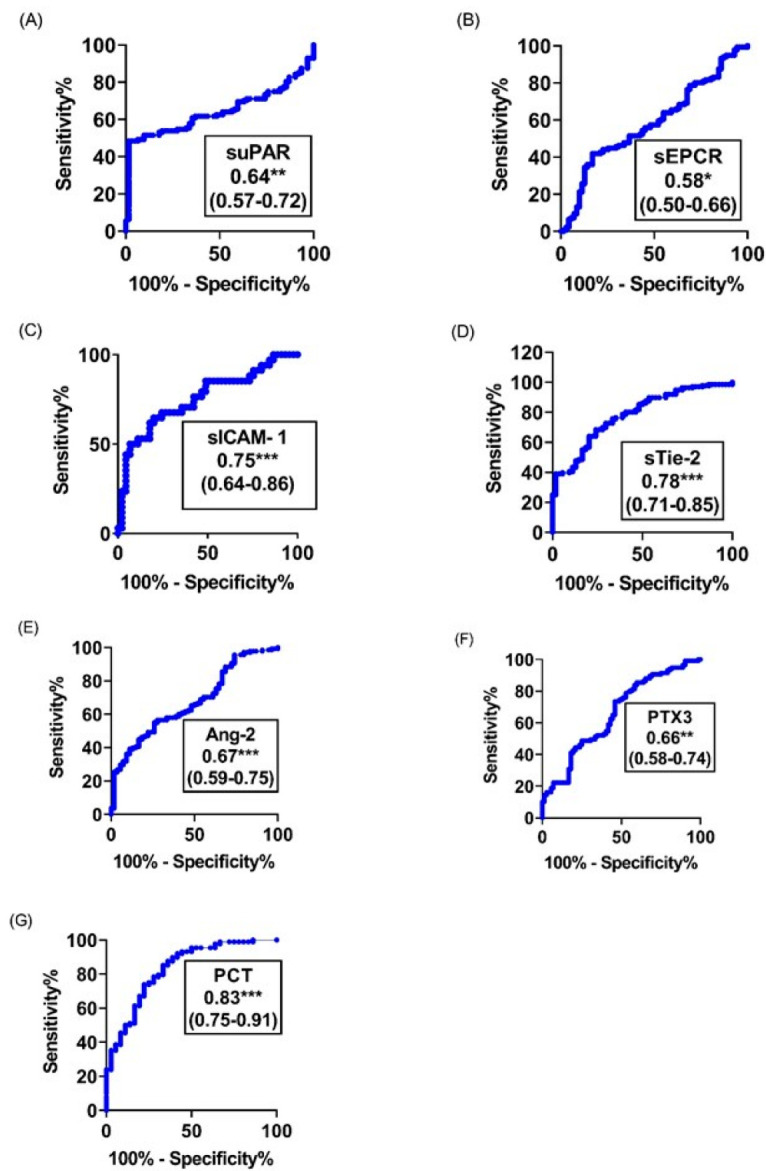
Assessment by ROC curve analyses of the individual prediction performance of PCT, sTie-2, Ang-2, PTX3, suPAR, sEPCR and sICAM-1 to differentiate severe from non-severe malaria cases. (**A**–**G**), Assessment by ROC curve analyses of the individual prediction performance of (**A**) suPAR, (**B**) sEPCR, (**C**) sICAM-1, (**D**) sTie-2, (**E**) Ang-2, (**F**) PTX3 and (**G**) PCT, to differentiate severe from non-severe cases on the day of admission to hospital. The area under the curve as well as the 95% confidence intervals are indicated in the legend. The * indicates 0.01 < *p*-value < 0.05, ** 0.001 < *p* value < 0.01, *** indicates *p* value ≤ 0.001.

**Figure 5 diagnostics-12-00524-f005:**
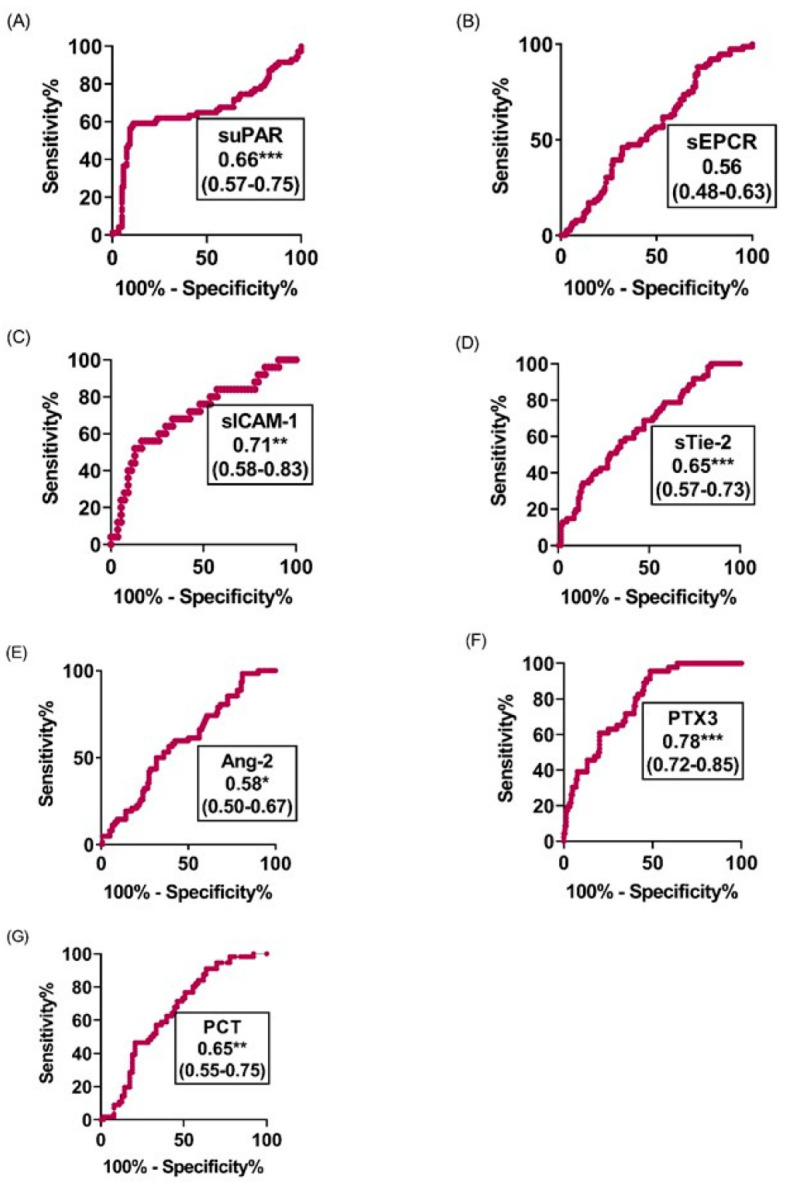
ROC curve analyses of the individual prediction performance of PTX3, sICAM-1, suPAR, sTie-2, PCT, Ang-2 and sEPCR to differentiate coma (BCS < 3) from non-coma (BCS > 2) among severe cases on the day of admission to hospital. (**A**–**G**), Assessment by ROC curve analyses of the individual prediction performance of (**A**) suPAR, (**B**) sEPCR, (**C**) sICAM-1, (**D**) sTie-2, (**E**) Ang-2, (**F**) PTX3 and (**G**) PCT to differentiate patients with deep coma cases (BCS < 3) from those without coma (BCS > 2) among severe cases diagnosed on the day of admission to hospital. The area under the curve as well as the confidence intervals are indicated in the legend. The * indicates 0.01 < *p* value < 0.05, ** 0.001 < *p* value < 0.01, *** indicates *p* value ≤ 0.001.

**Figure 6 diagnostics-12-00524-f006:**
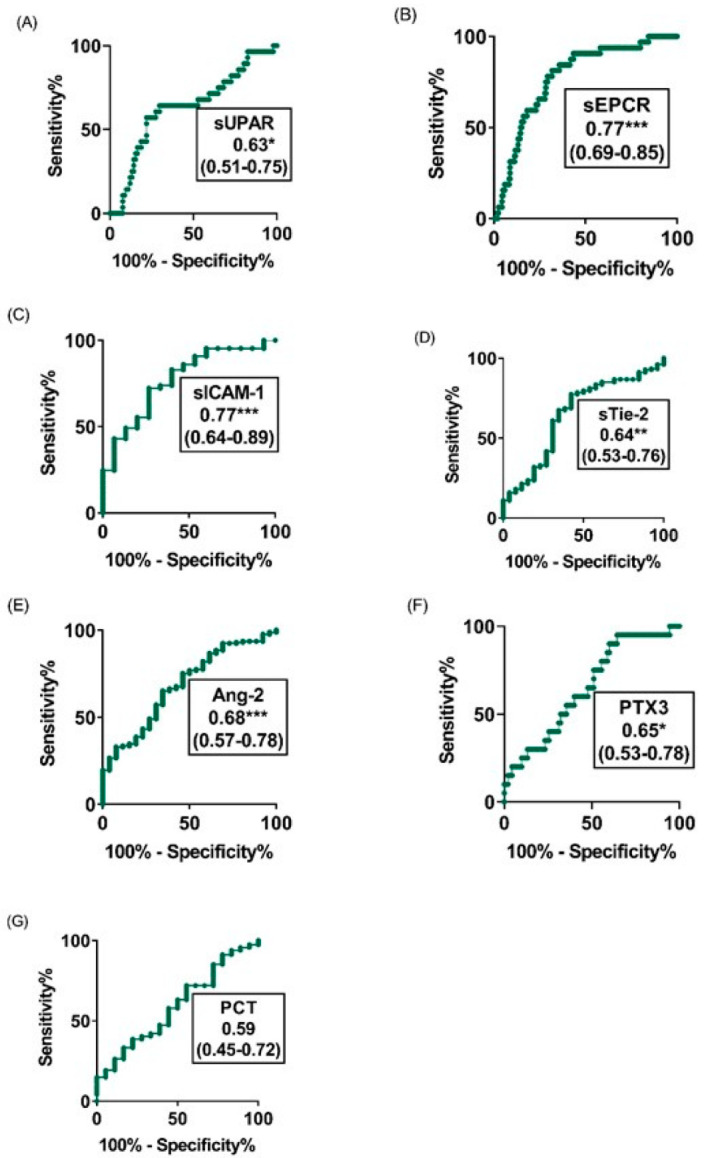
ROC curve analyses of the individual prediction performance of sEPCR, sICAM-1, sTie-2, Ang-2, PTX3, suPAR and PCT to differentiate fatal from non-fatal cases on the day of admission to hospital. (**A**–**G**), Assessment by ROC curve analyses of the individual prediction performance of (**A**) suPAR, (**B**) sEPCR, (**C**) sICAM-1, (**D**) sTie-2, (**E**) Ang-2, (**F**) PTX3 and (**G**) PCT to differentiate, on the day of admission to hospital, patients who will survive from those who will subsequently die. The area under the curve as well as the confidence intervals are indicated in the legend. The * indicates 0.01 < *p* value < 0.05, ** 0.001 < *p* value < 0.01, *** indicates *p* value ≤ 0.001.

**Figure 7 diagnostics-12-00524-f007:**
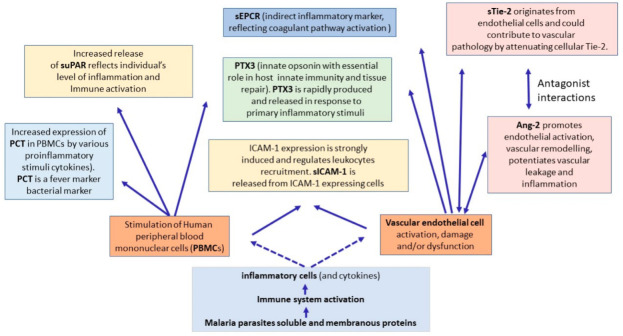
Clustering diagram showing the interaction between the molecules assessed in this study in the context of malaria pathogenesis.

**Table 1 diagnostics-12-00524-t001:** Biological, clinical and parasitological characteristics of the Beninese children enrolled in the study at admission. Kruskal–Wallis tests in conjunction with Dunn’s multiple comparison post-hoc tests were used for comparison between the clinical groups.

	Severe Cerebral Malaria (CM) (*n* = 101)	Severe Non-Cerebral Malaria (SNCM) (*n* = 133)	Uncomplicated Malaria (UM) (*n* = 105)	*p*-Value
Age (months), [IQR]	30 [5–60]	36 [4–60]	36 [5–66]	
Sex ratio (female/male)	45/56	72/61	48/57	
Temperature, median [IQR]	38.7 [36.3–41.4]	38,3 [36.5–41.5]	38.5 [36–40.8]	
Parasitaemia (P/μL), median [IQR]	44,000 [240–196,875]	65,457 [275–349,650]	64,533.5 [218–992,000]	
Haemoglobin (g/dL), median [IQR]	5.2 [2.3–12.9]	1.92 [0.6–12.8]	9.215 [5–15.1]	<0.0001
Blantyre coma score, median [IQR]	[0–2]	[3–5]	5	<0.0001
Severe malaria anemia (%)	39 (41%)	53 (39.84%)	0	<0.0001
Number of deaths	29	11	0	0.0005

**Table 2 diagnostics-12-00524-t002:** Relationship between suPAR, sEPCR, sICAM-1, sTie-2, Ang-2, PTX3 and PCT levels and malaria outcomes. Odds ratios for various outcomes when testing 7 analytes as potential predictors of severity, coma and mortality.

	Severity: CM/SM versus UM		Coma: BCS < 3 versus BCS > 2 among SM and CM Group		Mortality: Survivors versus Dead	
Molecules	OR (95% CI)	*p*-Value	OR (95% CI)	*p*-Value	OR (95% CI)	*p*-Value
suPAR	1.06 (1.03.1.09)	<0.001	1.04 (1.02.1.07)	<0.001	1.03 (1.01.1.05)	0.014
sEPCR	1.00 (1.00.1.00)	0.461	1.00 (1.00.1.00)	0.648	1.02(1.0.1.03)	0.003
sICAM-1	1.01 (1.00.1.01)	<0.001	1.00 (1.00.1.01)	0.004	1.0 (1.00.1.01)	0.199
sTie-2	1.21 (1.13.1.30)	<0.001	1.08 (1.03.1.14)	0.001	1.04 (0.97.1.12)	0.314
Ang-2	1.51 (1.17.1.94)	0.002	1.05 (0.94.1.19)	0.353	1.08 (0.94.1.23)	0.302
PTX 3	1.03 (1.01.1.05)	0.002	1.04 (1.03.1.06)	<0.001	1.02 (1.00.1.04)	0.0453
PCT	1.04 (1.02.1.07)	<0.001	1.00 (1.00.1.00)	0.975	1.00 (1.00.1.00)	0.75

**Table 3 diagnostics-12-00524-t003:** The three most predictive biomarkers of severity, coma and mortality are classified according to the AUC results. (AUC values are indicated between brackets as well as some of the main functional characteristics of the molecules).

Severity	Coma	Death
PCT (0.83) (acute phase reactant increased by inflammation)	PTX3 (0.78) (inflammation-angiogenesis regulator)	sEPCR (0.77) (increased by inflammation)
sTie-2 (0.78) (angiogenesis control)	sICAM-1 (0.71) (inflammation, tissue damage)	sICAM-1 (0.77) (inflammation, tissue damage)
sICAM-1 (0.75) (inflammation, tissue damage)	suPAR (0.66) (inflammation-angiogenesis)	Ang-2 (0.68) (angiogenesis activator)

## Data Availability

The datasets analyzed during the current study are not publicly available but can be provided by the corresponding author on reasonable request.

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
