# Peer review of "Specific Combinations of Inflammatory, Angiogenesis and Vascular Integrity Biomarkers Are Associated with Clinical Severity, Coma and Mortality in Beninese Children with Plasmodium Falciparum Malaria"

_diagnostics, 2022, doi:10.3390/diagnostics12020524_

Round 1
Reviewer 1 Report
Interesting results on Beninese children infect with malaria. Study design is well planned and the results are reasonable.
1. The title is abit too long and confusing
2. Label for A, B, C, on page 237 should be on the left side of the diagram
3. There is a missing term on page 6, line 167
4. Put in line for Table 1
5. Figure 3. The term survivors, survived. Please choose one.
6. Page 13, line 245, p-value
7. Use clustering diagram, to link up the possible markers.
8. Malaria-related deaths could be prevented if powerful diagnostic and reliable prognostic 22
biomarkers were available to allow rapid prediction of the clinical severity.
Comment about the recent emerging on molecular and hemozin-based approach such as
a. J. Han, “Micromagnetic resonance relaxometry for rapid label-free malaria diagnosis,” Nature medicine 20, 1069–1073 (2014).
b. “A lab-on-chip tool for rapid, quantitative, and stage-selective diagnosis of malaria,” Advanced Science , 2004101 (2021)
c. TP Loh, npj Aging and Mechanisms of Disease 6 (1), 1-12
d. TP Loh, Communications Biology 3 (1), 1-10
e. Scientific reports 3, 1–10 (2013)
f. Engineering Reports, e12383
Author Response
We thank the reviewer for the constructing comments on our manuscript and adress our responses that we do hope are appropriate. (comments are in italic and responses in green)
- The title is abit too long and confusing
We wanted to give the maximum information on the results of our study this is why the title is too long however, we propose the following title which is four words shorter and present the same information.
title : Association of vascular integrity, angiogenesis and inflammatory biomarkers with severity, coma and mortality in Beninese children with Plasmodium falciparum malaria
- Label for A, B, C, on page 237 should be on the left side of the diagram
We changed the A, B, C, to the left side of the diagram
- There is a missing term on page 6, line 167
We removed the space introduced by mistake that makes the sentence look like a word is messing.The following sentence is complete:
However, except the level of sEPCR, the concentrations of the other molecules were also significantly increased in children with CM as defined by coma with a BCS <3 compared to those among children with SNCM as defined by a BCS>2.
- Put in line for Table 1
Lines were added in table 1
- Figure 3. The term survivors, survived. Please choose one.
We choose the term survived to use in all graphic of figure 3
- Page 13, line 245, p-value
We corrected p value to p-value
- Use clustering diagram, to link up the possible markers.
We added a clustering diagram to show the known interaction of the studied molecules
- Malaria-related deaths could be prevented if powerful diagnostic and reliable prognostic biomarkers were available to allow rapid prediction of the clinical severity.
We did not comment on the new diagnosis tools based on magnetic resonance relaxometry (MRR) in this manuscript as the issue of our research is to prospect for markers that can differentiate between patients with distinct clinical manifestations of malaria and not to diagnose malaria infection in itself. Actually MRR is an original and innovative method which seems to be powerful to detect very low number of parasites within few minutes with accuracy. This may improve malaria diagnosis for clinical purposes as well as for epidemiological surveys. There is still to demonstrate its applicability in the field and its added value compared to rapid diagnostic tests (RDT) based on parasitic proteins HRP and LDH. It is very likely that false negatives by HRP-based RDT due to deletions in this protein will be detected by this method.

Reviewer 2 Report
This manuscript, "Specific combinations of inflammatory, angiogenesis and vascular integrity biomarkers are associated with clinical severity, coma and
mortality in Beninese children with Plasmodium falciparum malaria," by Tornyigah et al. is appropriate for publication in Diagnostics, in the opinion of this reviewer.
The only real concerns that this reviewer has about this manuscript are that the groups are relatively non-discriminating (no differentiation of male vs female, for example), and combining the data for cerebral malaria and severe, non-cerebral malaria. These decisions were perhaps due to the relatively small sample size, which presumably did not give differentiation between such groupings, as well as loosing statistical significance if the groups were broken-out more. Still, the results are interesting, and may prove useful to other researchers trying to pursue related studies. Likewise for age (although there is a statement at line 129 about this); there are reasons to think that adult CM vs pediatric CM are not exactly the same thing.
One final issue for the Editorial staff to consider is whether the very cursory coverage of Human Subjects Approval, monitoring, etc. are covered sufficiently by the statement at lines 411 & 412, "Informed Consent Statement: For all the participants a signed informed consent was obtained from parents or legal guardians." This reviewer is not entirely comfortable with only this statement.
Author Response
We thank the reviewer for the constructing comments on our manuscript and adress our responses that we do hope are appropriate. (comments are in italic and responses are in green)
1)The only real concerns that this reviewer has about this manuscript are that the groups are relatively non-discriminating (no differentiation of male vs female, for example), and combining the data for cerebral malaria and severe, non-cerebral malaria. These decisions were perhaps due to the relatively small sample size, which presumably did not give differentiation between such groupings, as well as loosing statistical significance if the groups were broken-out more. Still, the results are interesting, and may prove useful to other researchers trying to pursue related studies. Likewise for age (although there is a statement at line 129 about this); there are reasons to think that adult CM vs pediatric CM are not exactly the same thing.
1) In our logistical regression analysis we adjusted according to sex and age parasiteamia and other confounding factors. We are absolutely convinced that pediatric and adult cerebral are different. However, our research target pediatric cerebral malaria, our cohort is exclusively comped of children less than 5 years old and there were no significant difference in age between the clinical groups.
2) One final issue for the Editorial staff to consider is whether the very cursory coverage of Human Subjects Approval, monitoring, etc. are covered sufficiently by the statement at lines 411 & 412, "Informed Consent Statement: For all the participants a signed informed consent was obtained from parents or legal guardians." This reviewer is not entirely comfortable with only this statement.
2) Our research protocol was extensively reviewed by the Comité National d’Ethique pour la Recherche en Santé (CNERS), Cotonou, Benin (No 50, 25th October 2017, IRB00006860). This committee approved the study protocol and the informed consent that we presented to parents and guardians for agreement. We added the following statement: Informed Consent Statement: For all the participants a signed informed consent was obtained from parents or legal guardians, after full explanation of the study in their mother tongue by native speaking assistant.
The manuscript was corrected by an native english speaking scientist for language style.
Reviewer 3 Report
Authors analysed seven biomarkers Procalcitonin, Pentraxine-3, Ang-2, sTie-2, suPAR, sEPCR, and sICAM-1 in association with clinical malaria severity, coma and mortality in Beninese children. All markers were increased during severe or cerebral malaria compared to uncomplicated malaria. Pentraxine-3, Procalcitonin, suPAR, sTie-2, sICAM-1 and Ang-2 were significantly higher in children with coma signs. Pentraxine-3, suPAR, sEPCR, sICAM-1, sTie-2 and Ang-2 were higher among children who died from severe malaria compared to those who survived.
Minor points are:
- Ref [1]: update the WORLD MALARIA REPORT data from 2019 to 2021. Add the http link.
- The review regarding the fundamental premise for research is Ref [2] from 1993: please add more recent reference.
- Methods, line 105, specify the producer of the blood collecting tubes.
- Fig 1-3 are generally well-done. Just for better figure perception, the colours of experimental points could be changed to avoid too much similar actual black vs dark grey colours and all bars in black.
- Line 266, typo in Table number, it is Table 3.
- Discussion, line 357, some inaccuracies in the phrase “PTX3 is stored in specific granules in neutrophils and is released...”, because not only neutrophils are able to produce and release PTX3. Thus correct the statement with correspondent reference.
- Discussion, line 321, “…the enhanced level of Ang-2 and sTie-2 leads to sICAM-1 expression”. This is overstatement because no data is presented to support “leads” statement. Probably these are independent events, or causal relationship in studied patients could be supposed only. The citation of appropriate reference could support the supposition.
Author Response
Dear editor,
We thank the reviewer for the constructing comments on our manuscript and adress our responses that we do hope are appropriate.
- Ref [1]: update the WORLD MALARIA REPORT data from 2019 to 2021. Add the http link.
- We updated the information from the WORLD MALARIA REPORT 2021 and added the web link
- The review regarding the fundamental premise for research is Ref [2] from 1993: please add more recent reference.
2 We added a recent reference regarding the immunopathology of malaria
- Methods, line 105, specify the producer of the blood collecting tubes.
3 Blood collecting tubes were from BD Medical® ZI des Iles - BP4 - 38801 - Le Pont de Claix, France
- Fig 1-3 are generally well-done. Just for better figure perception, the colours of experimental points could be changed to avoid too much similar actual black vs dark grey colours and all bars in black.
4 We changed the black and grey and used other colors for better perception of experiment points.
- Line 266, typo in Table number, it is Table 3.
5 We changed the number from table 4 to table 3
- Discussion, line 357, some inaccuracies in the phrase “PTX3 is stored in specific granules in neutrophils and is released...”, because not only neutrophils are able to produce and release PTX3. Thus correct the statement with correspondent reference.
6 We corrected this inaccurate information and gave the corresponding reference
- Discussion, line 321, “…the enhanced level of Ang-2 and sTie-2 leads to sICAM-1 expression”. This is overstatement because no data is presented to support “leads” statement. Probably these are independent events, or causal relationship in studied patients could be supposed only. The citation of appropriate reference could support the supposition.
7 We changed this statement to “the enhanced level of Ang-2 and sTie-2 was concomitant with the expression of sICAM-1 expression”.